# A cross-sectional investigation of back pain beliefs and fear in physiotherapy and sport undergraduate students

**Cameron Black**[1], **Adrian Mallows**[2], **Sally Waterworth**[2], **Paul Freeman**[2], **Edward Hope**[2], **Bernard X. W. Liew**[2]*

**1** Occupational Health and Wellbeing, Buckinghamshire Healthcare NHS Trust, Stoke Mandeville Hospital, Aylesbury, United Kingdom, **2** School of Sport, Rehabilitation and Exercise Sciences, University of Essex, Colchester, Essex, United Kingdom

* bl19622@essex.ac.uk, liew_xwb@hotmail.com

**Data Availability Statement:** All relevant data are within the paper and its Supporting Information files.

## Abstract

### Background

Although low back pain (LBP) beliefs have been well investigated in mainstream healthcare discipline students, the beliefs within sports-related study students, such as Sport and Exercise Science (SES), Sports Therapy (ST), and Sport Performance and Coaching (SPC) programmes have yet to be explored. This study aims to understand any differences in the beliefs and fear associated with movement in students enrolled in four undergraduate study programmes–physiotherapy (PT), ST, SES, and SPC.

### Method

136 undergraduate students completed an online survey. All participants completed the Tampa Scale of Kinesiophobia (TSK) and Back Beliefs Questionnaire (BBQ). Two sets of two-way between-subjects Analysis of Variance (ANOVA) were conducted for each outcome of TSK and BBQ, with the independent variables of the study programme, study year (1st, 2nd, 3rd), and their interaction.

### Results

There was a significant interaction between study programme and year for TSK ($F_{(6, 124)}$ = 4.90, $P < 0.001$) and BBQ ($F_{(6, 124)}$ = 8.18, $P < 0.001$). Post-hoc analysis revealed that both PT and ST students had lower TSK and higher BBQ scores than SES and SPC students particularly in the 3rd year.

### Conclusions

The beliefs of clinicians and trainers managing LBP are known to transfer to patients, and more negative beliefs have been associated with greater disability. This is the first study to understand the beliefs about back pain in various sports study programmes, which is timely, given that the management of injured athletes typically involves a multidisciplinary team.

**Funding:** The author(s) received no specific funding for this work.

**Competing interests:** The authors have declared that no competing interests exist.

## Introduction

Low back pain (LBP) is the leading cause of years lived with disability globally [1], with high socio-economic cost [2], particularly among individuals with persistent symptoms [3]. The management of those with persistent, chronic low back pain (CLBP) is one of the most challenging aspects of clinical care [4]. Over the last decade, numerous clinical-practice guidelines (CPGs) have been published recommending the biopsychosocial model of care for CLBP [5, 6]. However, there has been an alarming lack of adherence to these recommendations [7–9], with the literature revealing a predominance of the biomedical model of care for CLBP [10].

The lack of adherence to LBP guidelines may be attributed to preconceived beliefs held by healthcare professionals (HCP) on the cause and optimal management of LBP [8]. The beliefs of HCP can explain as much as one-fifth of the variance in their recommendations to a patient with LBP [11]. There is consistent and strong evidence that HCP with a biomedical orientation or elevated fear-avoidance beliefs are more likely to advise patients to limit work and less likely to adhere to guidelines [12–15]. Evidence suggests that LBP beliefs are already present in the emerging HCP workforce [16], which suggests that educational factors play an important role in the formation of such beliefs. To date, beliefs related to LBP have been investigated in students enrolled in HCP disciplines such as physiotherapy (PT) [17–21], medicine [22–24], nursing [17, 19, 20], pharmacy [22], chiropractic [22], occupational therapy (OT) [19, 22], and midwifery [23].

It has been reported that PT students had more positive beliefs about LBP than medical, OT, and pharmacy students [22], as well as more positive beliefs about the harmfulness of common daily activities than OT and nursing students, respectively [19], but there was no differentiation among academic year groups. Understanding the effect of the study year and its possible interaction across different degree programmes may be important given research has reported an improvement in positive beliefs about LBP across study years [25]. Beliefs about LBP can also be affected by the course, with one study reporting that PT students had more positive LBP attitudes than non-healthcare students, although the nature of non-healthcare programmes were not provided [21].

Although LBP beliefs have been well investigated in mainstream HCP discipline students, the beliefs within sports-related study students, such as Sport and Exercise Science (SES), Sports Therapy (ST), and Sport Performance and Coaching (SPC) programmes have yet to be explored. This is surprising given that LBP [26, 27], fear and unhelpful beliefs are common amongst athletes, such as rowers [28, 29]. The management of athletes with LBP would often involve a multidisciplinary team that include professions beyond traditional HCP, such as sports scientists and coaches [30]. If different members of the multidisciplinary team have different beliefs about LBP, this can result in conflicting management and communication. Given that people with LBP want consistent information about their disorder, having different beliefs about LBP could potentially lead to suboptimal management of the disorder.

This study aims to understand any differences in the beliefs and fear associated with movement in students enrolled in four undergraduate study programmes–PT, ST, SES, and SPC, within a higher education institution (HEI) in the United Kingdom (UK). Prior research found that PT students have the most positive beliefs and that these can change over time based on new experiences and knowledge [20, 31]. Hence, we hypothesised that PT students have the most positive beliefs and the least amount of fear, when compared to the students enrolled on other programmes. Further, the effect of degree programmes will be greater in the latter study years.

## Methods

### Study design

This was a cross-sectional study involving an online survey (Qualtrics XM,Qualtrics, Provo, Utah, USA). Between November 2021 and April 2022, the survey was distributed to participants who were currently enrolled in either SES, ST, SPC or PT undergraduate programmes., which are all three-years in length. The PT programme is approved by the Health and Care Professions Council. Specific modules taught on each programme, along with a brief overview of their aims, study credits, and when teaching occurred, can be found in the supplementary material. Electronic informed consent was sought from all participants before study enrolment. Ethical approval was received from the University of Ethics human research ethics committee (ETH2122-0043).

### Power analysis

The *Superpower* package was used for power calculation [32]. A previous study reported mean Back Beliefs Questionnaire (BBQ) scores of 37.5, 35.3, 30.0 for PT, chiropractic, and pharmacy students, respectively [22]. Based on a one-way analysis of variance ANOVA with four levels, with a mean BBQ score of 38 (PT), 35 (ST), 30 (SES), and 30 (SPC); a standard deviation of 5 [22], a sample size of 30 participants in each group, would achieve a power of 0.97 at an alpha of 0.05.

### Participants

Participants were eligible for the study if they were enrolled in one of the four undergraduate courses (year 1, 2, or 3) of PT, SES, ST, and SPC degrees within the University for the academic year of 2021–2022. 166 undergraduate students participated in the present study, with 136 providing complete data to be included in the analysis. The descriptive characteristics of the included participants can be found in Table 1.

### Survey

**Tampa Scale of Kinesiophobia– 11 items (TSK-11).** The TSK-11 is an 11-item questionnaire [33]. Each item is scored on a 4-point Likert scale, ranging from 1 'strongly disagree' to 4

**Table 1. Descriptive characteristics of included participants.**

|  | Sport & Exercise Science | | | Sport & Performance Coaching | | | Sports Therapy | | | Physiotherapy | | |
|---|---|---|---|---|---|---|---|---|---|---|---|---|
| Variable | Yr 1, N = 13[1] | Yr 2, N = 26[1] | Yr 3, N = 9[1] | Yr 1, N = 12[1] | Yr 2, N = 14[1] | Yr 3, N = 10[1] | Yr 1, N = 11[1] | Yr 2, N = 7[1] | Yr 3, N = 10[1] | Yr 1, N = 8[1] | Yr 2, N = 6[1] | Yr 3, N = 10[1] |
| **Age** | 18.92 (0.64) | 20.31 (1.29) | 21.22 (1.09) | 18.83 (0.83) | 20.36 (0.84) | 21.40 (0.84) | 19.00 (0.63) | 20.71 (1.11) | 22.00 (1.83) | 19.25 (0.46) | 20.33 (0.52) | 21.50 (0.71) |
| **Body mass (kg)** | 76.92 (14.00) | 75.26 (12.63) | 81.11 (16.22) | 78.17 (20.68) | 73.77 (25.53) | 80.30 (3.86) | 73.09 (9.07) | 70.00 (7.19) | 72.90 (7.62) | 71.75 (7.25) | 73.33 (6.38) | 75.60 (11.27) |
| **Height (m)** | 1.73 (0.11) | 1.74 (0.09) | 1.79 (0.08) | 1.70 (0.11) | 1.72 (0.13) | 1.76 (0.05) | 1.70 (0.09) | 1.72 (0.07) | 1.75 (0.06) | 1.71 (0.10) | 1.70 (0.07) | 1.72 (0.09) |
| **Gender** |  |  |  |  |  |  |  |  |  |  |  |  |
| Male | 11 / 13 (85%) | 22 / 26 (85%) | 8 / 9 (89%) | 11 / 12 (92%) | 7 / 14 (50%) | 9 / 10 (90%) | 10 / 11 (91%) | 5 / 7 (71%) | 9 / 10 (90%) | 8 / 8 (100%) | 6 / 6 (100%) | 9 / 10 (90%) |
| Female | 2 / 13 (15%) | 4 / 26 (15%) | 1 / 9 (11%) | 1 / 12 (8.3%) | 7 / 14 (50%) | 1 / 10 (10%) | 1 / 11 (9.1%) | 2 / 7 (29%) | 1 / 10 (10%) | 0 / 8 (0%) | 0 / 6 (0%) | 1 / 10 (10%) |
| **LBP history (yes)** | 11 / 13 (85%) | 19 / 26 (73%) | 7 / 9 (78%) | 6 / 12 (50%) | 11 / 14 (79%) | 8 / 10 (80%) | 9 / 11 (82%) | 7 / 7 (100%) | 8 / 10 (80%) | 7 / 8 (88%) | 4 / 6 (67%) | 8 / 10 (80%) |

'strongly agree'; total scores vary between 11 and 44, with higher scores indicating higher levels of fear of movement-related pain [33]. The TSK has acceptable to excellent psychometric properties which have been previously reported [34].

**The Back Beliefs Questionnaire (BBQ).** The BBQ is a 14-item questionnaire, with five distractor items (questions 4, 5, 7, 9 and 11) that are not included in the final score [35]. Each item is scored on a 5-point Likert scale, ranging from 1 "strongly disagree" to 5 "strongly agree". Given that the remaining items are reverse scored, the total score ranges from 9 to 45, with higher scores indicating a more optimistic belief about the consequences of LBP. BBQ has acceptable psychometric properties, which have been previously reported [36].

### Statistical analysis

All analyses were performed in R software. For the dependent variables of TSK and BBQ score, two separate two-way between-subjects Analysis of Variance (ANOVA) were conducted, with the independent variables of the study programme (SES, SPC, ST, PT), study year (1st, 2nd, 3rd), and their interaction. Pairwise contrast via estimated marginal means was performed for post-hoc analysis where the primary analysis indicated a statistical significance. For all analyses, an alpha threshold of 0.05 was used to determine statistical significance.

### Results

For TSK score, there was a significant interaction between study programme and year ($F_{(6, 124)} = 4.90$, $P < 0.001$), and main effects of study programme ($F_{(3, 124)} = 9.80$, $P < 0.001$) and year ($F_{(2, 124)} = 78.54$, $P < 0.001$) (Fig 1). Between study programme differences were found only in the 3rd-year students. Post-hoc analysis revealed that ST students had lower TSK scores compared to SES and SPC students by 7.63 (95%CI 1.13 to 14.14, $P = 0.014$) and 11.40

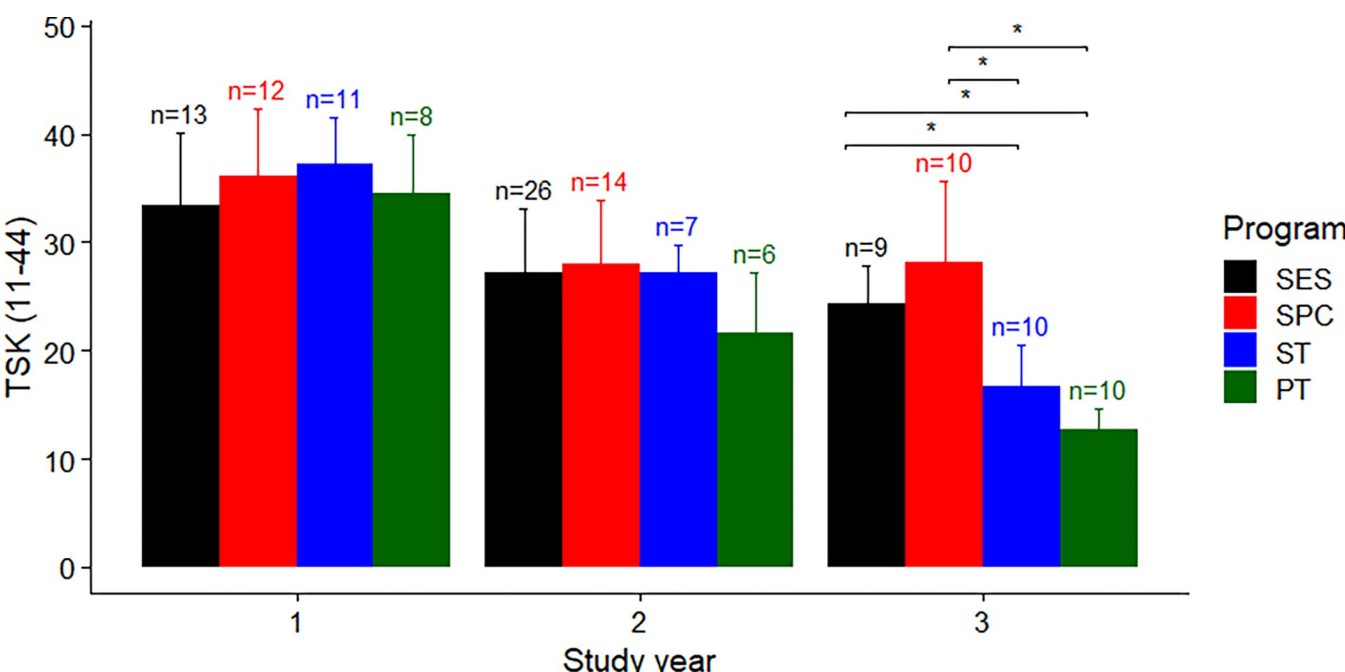

**Fig 1. Group average with error bars as one standard deviation of the total score of the Tampa Scale of Kinesiophobia 11-item version, for each study programme and year.** * Indicates a statistically significant ($P < 0.05$) pairwise difference. Abbreviation: SES–Sport and Exercise Science, SPC–Sport and Performance Coaching, ST–Sports Therapy, PT–Physiotherapy.

(95%CI 5.07 to 17.73, P < 0.001), respectively (Fig 1). In addition, PT students also had lower TSK scores compared to SES and SPC students by 11.63 (95%CI 5.13 to 18.14, P < 0.001) and 15.40 (95%CI 9.07 to 21.73, P < 0.001), respectively (Fig 1).

For BBQ score, there was a significant interaction between study programme and year (F(6, 124) = 8.18, P < 0.001), and main effects of study programme (F(3, 124) = 14.55, P < 0.001) and year (F(2, 124) = 64.19, P < 0.001) (Fig 2). In 3rd year students, post-hoc analysis revealed that SES students had greater BBQ score compared to SPC by 8.02 (95%CI 0.29 to 15.76, P = 0.039), and a lesser score compared to ST and PT students by 10.69 (95%CI 2.94 to 19.41, P = 0.003) and 14.98 (95%CI 7.24 to 22.71, P < 0.001), respectively (Fig 2). In 3rd year students, SPC students had lower BBQ score compared to ST and PT students by 18.70 (95%CI 11.17 to 26.23, P < 0.001) and 23.00 (95%CI 15.47 to 30.53, P < 0.001), respectively (Fig 2). In 2nd year students, SPC students had lower BBQ score compared to PT students by 9.00 (95%CI 0.79 to 17.21, P = 0.026) (Fig 2).

## Discussion

Negative beliefs and fear perceptions of LBP harboured during training have been suggested to play a critical role in influencing beliefs and emotions when entering the workforce [17]. This study set out to explore any differences in the beliefs and fear associated with movement in students enrolled in four undergraduate study programmes–PT, ST, SES, and SPC, within a HEI in the UK. In partial support of our hypotheses, PT and ST students had greater positive beliefs and lower levels of fear than SES and SPC students, which largely occurred in the third year of studies. The similar trajectory of PT and ST students' beliefs and fear compared to the other cohorts suggest there may be an important distinction in their educational experiences compared to SES and SPC study programmes. Given that the management of LBP within the

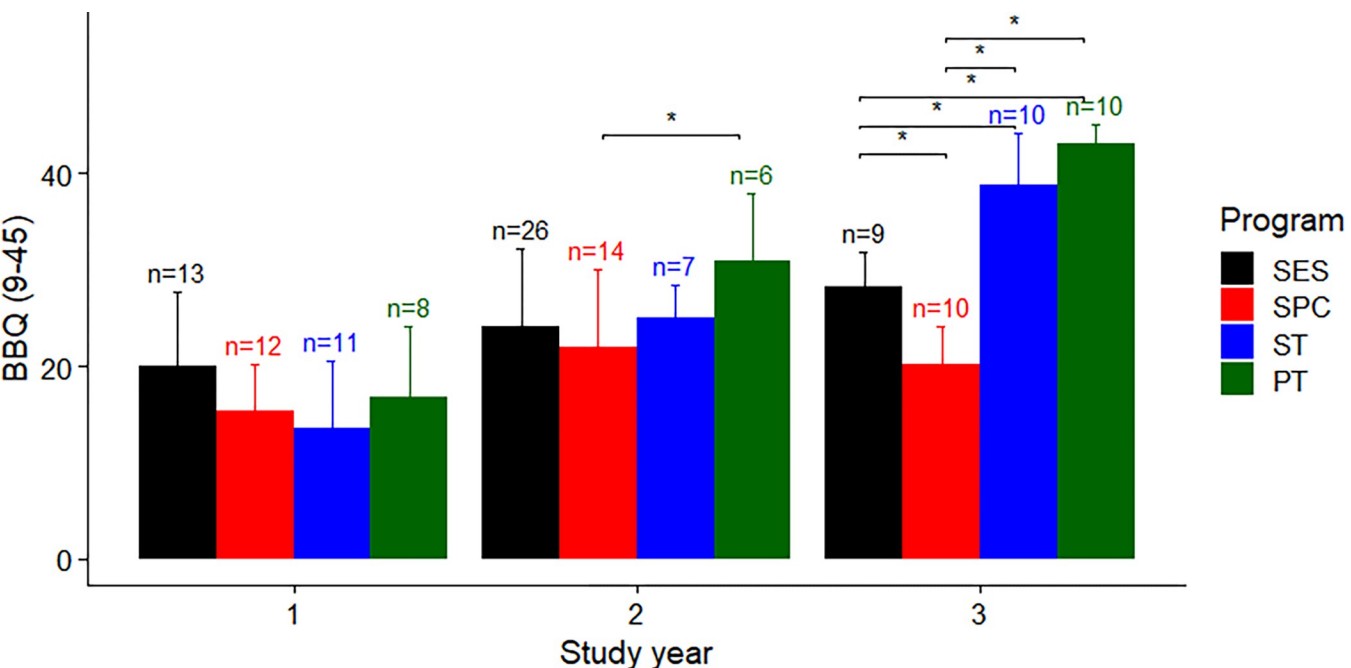

**Fig 2. Group average with error bars as one standard deviation of the total score of the Tampa Scale of Kinesiophobia 11-item version, for each study programme and year.** * Indicates a statistically significant (P < 0.05) pairwise difference. Abbreviation: SES–Sport and Exercise Science, SPC–Sport and Performance Coaching, ST–Sports Therapy, PT–Physiotherapy.

sporting environment relies on multidisciplinary care, our findings provide the basis upon which management might differ between professional disciplines and the contributing factors of these differences.

No studies to our knowledge have compared LBP beliefs and fear in PT students with those enrolled in various sports-study programmes. A recent systematic review of back beliefs suggested that a sum score above 27 indicated positive beliefs about the BBQ [37]. It is interesting to note that PT and ST students scored well above this cut-off score and well above most of the studies included in the review (8 out of 12 studies). One study reported that final-year PT and chiropractic students had more helpful LBP beliefs than OT, medicine, and pharmacy students [22], which indirectly supports the finding of the present study. Comparing the BBQ score, 3rd year PT and ST students reported a mean score of 43.2 and 38.9 in the present study, whilst final-year PT and chiropractic students in another study reported a mean score of 37.5 and 35.3, respectively [22]. One study reported a mean BBQ score of 33.7 across study years in PT students, but scores for each year were not reported [23]. Kennedy et al. (2014) [23] also reported that PT students had more positive beliefs toward LBP than medical and nursing students, and such beliefs significantly improved over their study years.

The presence of more positive back pain beliefs and lower fear levels in 3rd year PT and ST students, compared to SES and SPC students, could be due to several factors. These include curricula differences within each programme, including specific contents of taught modules and the presence of mandatory clinical placements in PT and ST. The more positive beliefs and lower fear levels could be due to the embedment of (spinal) pain education into the curricula of PT and ST students. A previous study reported that chiropractic and PT students received a total of 310 hrs and 112.5 hrs of curricula related to the management of spinal pain in their final year, compared to medicine (4 hrs, BBQ = 32.6), OT (10 hrs, BBQ = 31.8), and pharmacy students (2 hrs, BBQ = 30.0) [22]. Although a correlational analysis was not conducted, speculatively, the relationship between the number of hours of curricula dedicated to spinal pain and BBQ score could be non-linear [22]. A previous study reported that short (6.5h) education sessions focusing on spinal pain mechanisms can promote more positive back pain beliefs and attitudes in HCP [38]. This suggests that there is an optimal threshold of taught hours dedicated to spinal pain for an effect on BBQ score.

Having clinical experience with real-world patients experiencing LBP during clinical placements could be an important contributor to back pain beliefs. This could explain why PT students had more positive beliefs in their 2nd year, when they had already started clinical placements, compared to ST students who only do such placements in their final year. The present cohort of 2nd year PT students had a mean BBQ score of 31.0, which was comparable with PT students (BBQ = 30.7) enrolled in HEIs in Australia, Taiwan, and Singapore [39]. In a qualitative study of medical students, learning from clinical placements was cited as an important source of influence for their back pain beliefs [40]. Presently, it is uncertain what elements within clinical placements—such as patient interaction, case discussion with a HCP, etc, drive an improvement in back pain beliefs. Yet, as some HCPS have been shown to display negative back pain beliefs [12], this may conflict with the learning of students when forming beliefs about LBP.

The more negative beliefs and greater fear levels in SES and SPC students provide opportunities for curricula modification to prepare them professionally for their encounter with athletes suffering from LBP. Although it is not realistic to expect SES and SPC students to receive extensive healthcare training, embedding short education sessions focusing on spinal or general musculoskeletal pain education could be incorporated into exercise or rehabilitation-focused modules. Given that sports-related study programmes within the UK HEI setting typically co-exist within the same faculty, opportunities for cross-disciplinary teaching may be

useful in knowledge exchange between HCPs and sports practitioners. From an applied perspective, and given the prevalence of LBP amongst athletes [26, 27], a greater understanding of the condition will facilitate multidisciplinary management of said individuals [30].

## Limitations

This study has some limitations. We did not account for the potential confounding effect of LBP history and sex in our analysis, as it did not form our primary hypothesis, and would have substantively increased the same size requirement. However, previous studies reported no statistically significant effect of LBP history of back pain beliefs [18–20]. There is some evidence that female participants report greater levels of fear to pain than male participants [41]. It is unlikely that the lower BBQ and TSK scores reported in PT students was attributed to a lack of female participants in the group. This was because the lack of female participants in the PT group was present across all year groups, but significance was detected largely in the 3rd year group. The cross-sectional nature precludes making within-subject inferences about whether the back pain beliefs and fear levels of a student improve across study years [25]. Given that the study programmes investigated in this study may differ in their curricula across different HEIs, extrapolating our findings nationally and internationally should be done with caution. A previous study reported the importance of the cultural context of higher education on back pain beliefs in physiotherapy students [39]. Even within the same country, similar study programmes between different HEIs may vary in the substantive content of their taught modules. For PT programmes, taught courses across different HEIs within the UK may be more easily comparable given they have to meet a common standard of proficiency set out by their regulatory body, although there may be some differences in how they achieve this.

## Conclusion

The effect of different undergraduate health and sport study programs on back pain beliefs and fear depended on the study year. PT and ST students have more positive back pain beliefs and lower fear levels than SES and SPC students, particularly when in their final (3rd) year. The beliefs of HCPs are known to transfer to patients, and more negative beliefs have been associated with greater disability. This is the first study to understand the beliefs about back pain in various sports study programmes, which is timely, given that the management of injured athletes typically involves a multidisciplinary team. Future research is required to investigate the elements within taught modules and clinical placements that contribute to the formation of back pain beliefs in students. This is critical to identify flexible and feasible strategies to improve the understanding of pain management across a heterogeneous set of study programmes.

## Supporting information

**S1 Data.**
(XLSX)

**S2 Data. Modules undertaken by the four study programs of physiotherapy, sports therapy, sports and exercise science, and sports performance and coaching in the 2021/22 academic year.**
(XLSX)

## Acknowledgments

We like to thank Adam Webster, Saif Siddiqi, and William Rodrigues Da Silva for assisting in data collection.

## Author Contributions

**Conceptualization:** Adrian Mallows, Bernard X. W. Liew.

**Data curation:** Bernard X. W. Liew.

**Formal analysis:** Cameron Black, Bernard X. W. Liew.

**Methodology:** Adrian Mallows, Bernard X. W. Liew.

**Project administration:** Sally Waterworth, Paul Freeman, Edward Hope, Bernard X. W. Liew.

**Software:** Bernard X. W. Liew.

**Supervision:** Sally Waterworth, Paul Freeman, Edward Hope, Bernard X. W. Liew.

**Validation:** Adrian Mallows, Sally Waterworth, Paul Freeman, Edward Hope, Bernard X. W. Liew.

**Visualization:** Bernard X. W. Liew.

**Writing – original draft:** Cameron Black, Adrian Mallows, Sally Waterworth, Paul Freeman, Edward Hope, Bernard X. W. Liew.

**Writing – review & editing:** Cameron Black, Adrian Mallows, Sally Waterworth, Paul Freeman, Edward Hope, Bernard X. W. Liew.

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
