## [Decision Letter · Decision Letter 0]

8 Jan 2023

PONE-D-22-34229Back pain beliefs and fear in physiotherapy and sport undergraduate studentsPLOS ONE

Dear Dr. Liew,

Thank you for submitting your manuscript to PLOS ONE. After careful consideration, we feel that it has merit but does not fully meet PLOS ONE’s publication criteria as it currently stands. Therefore, we invite you to submit a revised version of the manuscript that addresses the points raised during the review process.

We look forward to receiving your revised manuscript.

Kind regards,

Ravi Shankar Yerragonda Reddy, Ph.D

Academic Editor

PLOS ONE

Journal Requirements:

3. Please amend your manuscript to include your abstract after the title page. 

Reviewers' comments:

Reviewer's Responses to Questions

**Comments to the Author**

1. Is the manuscript technically sound, and do the data support the conclusions?

Reviewer #1: Yes

Reviewer #2: Partly

2. Has the statistical analysis been performed appropriately and rigorously? 

Reviewer #1: Yes

Reviewer #2: Yes

3. Have the authors made all data underlying the findings in their manuscript fully available?

Reviewer #1: Yes

Reviewer #2: No

4. Is the manuscript presented in an intelligible fashion and written in standard English?

Reviewer #1: Yes

Reviewer #2: Yes

5. Review Comments to the Author

Reviewer #1: Review comments on Manuscript Number: PONE-D-22-34229. Entitled "Back pain beliefs and fear in physiotherapy and sport undergraduate students"

Overall, the idea of research is very interesting to be studied nowadays and paper is coherently developed. However, there are some comments and suggestions.

Abstract

- Well structured

- Keywords: write it in alphabetical order

Introduction

- The abbreviations already mentioned in the abstract so no need to re-write complete terms

Materials and methods

- Well structured

Statistical analysis

- Well structured

References

Update some references is required

Reviewer #2: The manuscript title should include the type of the study. Conclusion of the study doesn't declare the end result of the study obviously, I think it should be re-written. The author should declare participants inclusion criteria rather than study programmes only.

6. PLOS authors have the option to publish the peer review history of their article (what does this mean?). If published, this will include your full peer review and any attached files.

Reviewer #1: No

Reviewer #2: No

While revising your submission, please upload your figure files to the Preflight Analysis and Conversion Engine (PACE) digital diagnostic tool, https://pacev2.apexcovantage.com/. PACE helps ensure that figures meet PLOS requirements. To use PACE, you must first register as a user. Registration is free. Then, login and navigate to the UPLOAD tab, where you will find detailed instructions on how to use the tool. If you encounter any issues or have any questions when using PACE, please email PLOS at figures@plos.org. Please note that Supporting Information files do not need this step.<quillbot-extension-portal></quillbot-extension-portal>

---

## [Author Response · Author response to Decision Letter 0]

24 Jan 2023

Please see uploaded document for a properly formatted response.

Reviewer #1

Review comments on Manuscript Number: PONE-D-22-34229. Entitled "Back pain beliefs and fear in physiotherapy and sport undergraduate students". Overall, the idea of research is very interesting to be studied nowadays and paper is coherently developed. However, there are some comments and suggestions.

Abstract

- Well structured

Reply: We thank the Reviewer for the positive comments. We will address all feedback and suggestions provided by the Reviewer.

- Keywords: write it in alphabetical order

Reply: We have reordered it to alphabetical order.

Introduction

- The abbreviations already mentioned in the abstract so no need to re-write complete terms

Reply: We thank the Reviewer for this comment. We kindly have to disagree with this feedback, as we feel that similar to Figures and Tables, the main manuscript should be worded independently from the Abstract. We have thus retained the definition of the abbreviations in the main manuscript. However, we will seek the guidance of the Editor on this matter too, during the resubmission process.

Materials and methods

- Well structured

Statistical analysis

- Well structured

Reply: We thank the Reviewer for the positive comments.

References

Update some references is required

Reply: We thank the Reviewer for this comment. We are unsure if the Reviewer is suggesting we ensure our References are follow PLoS’s guidance, or if new references are needed. We have ensured that our Reference format follows PLoS. We believe that we have tried to cite the latest evidence where available. We are very open to amending any references that the Reviewer thinks needs editing. We hope that the Reviewer can provide the relevant citations for us to consider including.  

Reviewer #2

The manuscript title should include the type of the study. 

Reply: We have modified the title to read as “A cross-sectional investigation of back pain beliefs and fear in physiotherapy and sport undergraduate students”.

 Conclusion of the study doesn't declare the end result of the study obviously, I think it should be re-written. 

Reply: We have added a sentence at the start of the Conclusion, that reads as:

The effect of different undergraduate health and sport study programs on back pain beliefs and fear depended on the study year.

The author should declare participants inclusion criteria rather than study programmes only.

Reply: We thank the Reviewer for this comment. We followed the broad inclusion criteria of prior similar publications (Inman and Ellard, 2022; Leysen et al., 2021), where the priminary inclusion criteria was that students are enrolled into the program. Given that the aim of the present study was to investigate the beliefs and fear in a representative sample of students in our defined study programs, there was no further inclusion criteria.

References

1.Inman, J.G.K., Ellard, D.R., 2022. What influences graduate medical students’ beliefs of lower back pain? A mixed methods cross sectional study. BMC Med Educ 22, 633.

2.Leysen, M., Nijs, J., Van Wilgen, P., Demoulin, C., Dankaerts, W., Danneels, L., . . . Roussel, N., 2021. Attitudes and beliefs on low back pain in physical therapy education: A cross-sectional study. Braz J Phys Ther 25, 319-28.

---

## [Decision Letter · Decision Letter 1]

10 Apr 2023

A cross-sectional investigation of back pain beliefs and fear in physiotherapy and sport undergraduate students

PONE-D-22-34229R1

Dear Dr. Bernard X W Liew,

We’re pleased to inform you that your manuscript has been judged scientifically suitable for publication and will be formally accepted for publication once it meets all outstanding technical requirements.

Kind regards,

Ravi Shankar Yerragonda Reddy, Ph.D

Academic Editor

PLOS ONE

Reviewers' comments:

Reviewer's Responses to Questions

**Comments to the Author**

1. If the authors have adequately addressed your comments raised in a previous round of review and you feel that this manuscript is now acceptable for publication, you may indicate that here to bypass the “Comments to the Author” section, enter your conflict of interest statement in the “Confidential to Editor” section, and submit your "Accept" recommendation.

Reviewer #1: All comments have been addressed

Reviewer #2: All comments have been addressed

2. Is the manuscript technically sound, and do the data support the conclusions?

Reviewer #1: Yes

Reviewer #2: Yes

3. Has the statistical analysis been performed appropriately and rigorously? 

Reviewer #1: Yes

Reviewer #2: Yes

4. Have the authors made all data underlying the findings in their manuscript fully available?

Reviewer #1: Yes

Reviewer #2: Yes

5. Is the manuscript presented in an intelligible fashion and written in standard English?

Reviewer #1: Yes

Reviewer #2: Yes

6. Review Comments to the Author

Reviewer #1: Review comments on Manuscript Number: PONE-D-22-34229R1. “Entitled "A cross-sectional investigation of back pain beliefs and fear in physiotherapy and sport undergraduate students"

Overall, the idea of research is very interesting, organized and well written reasonable. The authors have done great effort to accomplish this work. They fulfilled all reviewers' comments and made necessary changes throughput the manuscript.

Reviewer #2: The author has been addressed all previous comments and the manuscript seems to be corrected precisely

7. PLOS authors have the option to publish the peer review history of their article (what does this mean?). If published, this will include your full peer review and any attached files.

Reviewer #1: **Yes: **Amira M. Abd-elmonem

Reviewer #2: No

<quillbot-extension-portal></quillbot-extension-portal>

---

## [Editor Report · Acceptance letter]

12 Apr 2023

PONE-D-22-34229R1 

A cross-sectional investigation of back pain beliefs and fear in physiotherapy and sport undergraduate students 

Dear Dr. Liew:

I'm pleased to inform you that your manuscript has been deemed suitable for publication in PLOS ONE. Congratulations! Your manuscript is now with our production department. 

Kind regards, 

on behalf of

Dr. Ravi Shankar Yerragonda Reddy 

Academic Editor

PLOS ONE